# Animal Models Utilized for the Development of Influenza Virus Vaccines

**DOI:** 10.3390/vaccines9070787

**Published:** 2021-07-14

**Authors:** Ericka Kirkpatrick Roubidoux, Stacey Schultz-Cherry

**Affiliations:** Department of Infectious Diseases, St. Jude Children’s Research Hospital, Memphis, TN 38105, USA; ericka.roubidoux@stjude.org

**Keywords:** animal models, vaccines, influenza, ferret, mouse, guinea pig

## Abstract

Animal models have been an important tool for the development of influenza virus vaccines since the 1940s. Over the past 80 years, influenza virus vaccines have evolved into more complex formulations, including trivalent and quadrivalent inactivated vaccines, live-attenuated vaccines, and subunit vaccines. However, annual effectiveness data shows that current vaccines have varying levels of protection that range between 40–60% and must be reformulated every few years to combat antigenic drift. To address these issues, novel influenza virus vaccines are currently in development. These vaccines rely heavily on animal models to determine efficacy and immunogenicity. In this review, we describe seasonal and novel influenza virus vaccines and highlight important animal models used to develop them.

## 1. Introduction

Influenza viruses have caused seasonal epidemics and occasional pandemics for hundreds of years [1]. Influenza A viruses (IAV) were isolated in 1933 [2] and Influenza B viruses (IBV) were subsequently isolated in 1940 [3]. IAVs and IBVs are very diverse and rapidly evolve to evade immune responses, giving them the ability to cause millions of infections as well as thousands of hospitalizations and deaths annually [4,5]. Influenza viruses are categorized based on their two glycoproteins, hemagglutinin (HA) and neuraminidase (NA), into IAV Group 1 (H1, H2, H5, H6, H8, H9, H11, H12, H13, H16, N1, N4, N5 and N8), IAV Group 2 (H3, H4, H7, H10, H14, H15 N2, N3, N6, N7 and N9), IBV B/Yamagata-like lineage and IBV B/Victoria-like lineage. H1N1, H3N2 and both IBV lineage viruses infect humans seasonally [6]. IAVs can be found in many natural hosts including aquatic birds, domesticated poultry, swine, equines and canines [7,8,9,10]. Conversely, IBVs primarily infect humans and can be found sparingly in other mammalian hosts, such as the harbor seal [11,12].

Generally, seasonal influenza viruses cause mild to moderate infections in humans. Symptoms can include fever, headache, lethargy, loss of appetite and body aches. Occasionally, these infections are more serious and can lead to hospitalization or death [6]. However, pandemic influenza viruses cause much more severe infections and have significantly higher mortality rates. Pandemic IAVs have zoonotic origins and are introduced into the population by direct interaction with infected poultry, waterfowl or swine. The most notable influenza virus pandemic occurred in 1918 (H1N1 “Spanish Flu”) and caused over 21 million deaths world-wide [1]. The most recent influenza virus pandemic occurred in 2009 (H1N1 “Swine Flu”). This outbreak was less severe, causing 200,000 deaths [1]. Humans rarely transmit avian or swine viruses from person-to-person, however the threat of an influenza virus pandemic caused by a zoonotic virus that can transmit between humans is omnipresent. Constant global surveillance is critical to try and identify viruses with pandemic potential [4]. Previous outbreaks of avian H5N1, which can have mortality rate as high as 60%, were the motivation behind creating national stockpiles of seed stocks of pandemic potential influenza virus vaccines [13,14,15]. If an unexpected IAV subtype was to cause a pandemic, a matching vaccine would need to be developed, leading to a delay in vaccinations and possible high mortality in the population [16,17]. Vaccines are critical for the prevention of influenza virus infections. However, more “universal” influenza virus vaccines that could potentially protect from both epidemics and pandemics are needed. Animal models are the basis for many vaccine studies to establish efficacy, safety, and immunogenicity of a vaccine. This review will discuss the types of influenza virus vaccines under development and describe how animal models can be utilized to further their progress towards clinical trials.

## 2. Seasonal Influenza Virus Vaccines

The primary defense to protect from influenza virus infections are seasonal vaccines. These vaccines have varying efficacy, usually around 40–60% each year. However, during some influenza virus seasons, vaccine efficacy was found to be as low as 20% [4,18]. The first influenza virus vaccine produced was an inactivated form of A/Puerto Rico/8/1934 virus [19]. Monovalent vaccines were the first to be developed after the identification of the H1N1 subtype in humans [19]. Shortly after this, Influenza B virus was identified in humans, leading to a more comprehensive bivalent vaccine [20]. After the 1968 H3N2 pandemic, a trivalent vaccine was developed. This vaccine is composed of three influenza virus strains representing each subtype [1]. In the 1980s it was discovered that there are two co-circulating IBV lineages [21]. Another vaccine approach was adopted in the form of a quadrivalent vaccine which is composed of four representative strains, two for IAV and two for IBV [22,23,24,25].

There are several different vaccines currently on the market that fall into three categories- inactivated influenza vaccines (IIV), live-attenuated influenza vaccine (LAIV) and recombinant HA subunit vaccines (Table 1). These vaccines can be monovalent, trivalent (TIV) or quadrivalent (QIV). The HA of influenza virus is responsible for virus attachment and entry, meaning that antibody responses focused on the HA can prevent infection. Because of this, IIVs are standardized based on HA content rather than total protein in the hopes of generating a large, sterilizing immune response that could prevent both infection and transmission of viruses. LAIVs are administered based on an infectious dose, meaning that other viral proteins, such as the NA, are also included. The NA, and other proteins, are present in IIVs, however the amount of protein in a particular vaccine preparation can vary and is not standardized. IAV and IBV viruses are highly diverse, and both the HA and NA undergo rapid antigenic drift leading to the emergence of antigenically distinct variants every few years. To combat this, the vaccines must be reformulated annually. This puts pressure on vaccine manufacturers to constantly ensure that seasonal vaccines will be produced in time for the new influenza season [17,26].

In the northern hemisphere, the influenza season is from October to March with infections peaking between December and February [4]. About six months prior to the onset of each season, a group of experts comprised of representatives from the World Health Organization (WHO) Influenza Virus Collaborating Centers (WHO CCs), essential regulatory laboratories, and other partners, review data generated by the WHO Global Influenza Surveillance and Response System (GISRS) and make recommendations on influenza vaccine composition “seed strains” that will be the best candidates for vaccine production in the following season [17,28,29,30]. Production time from strain selection to vaccine distribution can take from six to eight months. Seed strains are selected based on several criteria including their ability to grow to high titers in embryonated hen eggs and their antigenic relatedness to prominent circulating strains [29]. Most seed strains are made as reassortant viruses (with internal genes from A/Puerto Rico/8/1934) to ensure that they will grow to high titers in a single egg [30]. Eggs are used extensively for propagating influenza viruses, making them a critical tool for virus and vaccine research [17,31]. Like any system, there are some drawbacks to using eggs as a tool for propagating influenza virus. Because avian species have different sialic acid receptors compared to humans, the viruses can acquire mutations that may alter antibody responses, drastically reducing vaccine effectiveness [32,33,34]. Approximately one vaccine dose can be produced per egg, meaning that millions of eggs are needed for any given influenza virus vaccine season. The possibility of egg shortages threatens production of influenza virus vaccines, both seasonal and pandemic. Additionally, production of H5N1 vaccines has been challenging in the past due to the high infectivity of avian viruses in embryonated eggs, which can lead to the death of the embryo before high virus titers are reached. This increases the number of eggs needed to prepare pandemic vaccines. To address these concerns and limitations, cell-culture based and recombinant protein based vaccines are available (Table 1) [17].

Typically, IIVs and recombinant HA vaccines are administered via a single intramuscular (IM) injection. LAIV vaccines are administered as a mist intranasally (IN). IIVs were originally made using whole inactivated virus. Whole virus IIVs could be very reactogenic, leading to the introduction of split-virion and subunit vaccines. Split vaccines are made using detergent to break open the viral particle, exposing both the internal and external proteins of the virus to the immune system. Subunit vaccines are then further purified to increase total HA content [35]. This method was shown to be just as effective, and less reactogenic, compared to whole virus vaccines. IIVs induce protective immune responses without the addition of an adjuvant. However, the use of adjuvants has been discussed to increase immunogenicity in the elderly instead of administering a high dose vaccine. Adding adjuvants could lead to the reduction of HA content in each dose, which could prevent vaccine shortages [17,26]. While they are not commonly used, there are a few adjuvants that are licensed for uses in influenza virus vaccines. Adjuvants have been demonstrated to boost immune responses, especially in high-risk groups that generally have suboptimal vaccine responses [17,36]. Alum salt was one of the first licensed adjuvants however newer adjuvants, such as MF59 and AS03, are oil-in-water emulsions. Many other adjuvants are currently being evaluated in animal models or have moved into clinical trials (detailed reviews in [17,36]). While improvements in seasonal vaccines could be helpful in curbing annual epidemics, they offer no protection against emerging pandemic influenza viruses. This constant threat has spurred interest into taking a different approach to influenza virus vaccines- going universal instead of strain-specific.

## 3. Universal Influenza Virus Vaccines

As discussed previously, the threat of pandemic influenza virus outbreaks, along with seasonal antigenic drift, has uncovered faults in current seasonal vaccines. A “universal” approach to influenza virus vaccines has become the primary focus of new strategies. Universal influenza virus vaccines must meet several requirements established by agencies including the WHO and the National Institute of Allergy and Infectious Diseases (NIAID) [37,38,39]. These requirements include producing vaccines that cause long-lived immune responses (one to five years), prevent clinical disease caused by IAV and IBV drifted subtypes, are safe for the general population and high-risk groups as well as provide protection from emerging pandemic strains.

The main premise for universal influenza virus vaccines is to boost antibody responses that target conserved, yet less accessible, regions of the virus [40,41,42,43,44,45,46,47]. The HA head domain is the primary target of immune responses after infection and vaccination. Due to the high variability of the HA head domain between strains circulating annually and between groups of IAV/IBV, it may not be the best target for universal vaccines. To overcome the immunodominance of the HA head domain, prime-boost methods and the use of adjuvants have been employed in hopes of diverting the immune response away from the HA head domain and focusing it more on conserved epitopes [48,49,50]. Additionally, several new approaches and targets have been proposed as universal influenza virus vaccines including; the use of recombinant proteins, virus-like particles (VLPs) [51], viral vectors [52,53,54], self-assembling nanoparticles [4,18] and nucleic acids (DNA [39] or mRNA [55,56,57,58]). A majority of recombinant proteins focus on targeting the HA stalk (chimeric HAs (cHAs) [59,60,61,62], stalk-only “headless HAs” [63] or mini-HAs [64]), conserved epitopes in both the head and stalk domains of the HA (mosaic HAs [65,66] and computational optimized broadly reactive antigen (COBRA) [67,68,69,70]), the neuraminidase [71,72,73,74], the matrix protein (M1) [43,75,76], the matrix 2 ectodomain (M2e) [75,76,77,78] and the nucleoprotein (NP) [79,80]. Many technologies have been designed to carry recombinant antigens in ways that boost immunogenicity or mimic natural infection. Virus-like particles have been engineered to carry recombinant antigens such as M2e [42] or recombinant HAs [81]. Viral vectors, such as the modified vaccina virus Ankara and chimpanzee adenovirus, have been engineered to stimulate T cell responses by using conserved epitopes on M1 and NP proteins [75,76,82]. Nanoparticle based vaccines have the advantage of being self-assembling structures that can mimic the structure of an actual virion; however, they are not infectious or self-replicating. They can be designed to express the full-length HA, HA stalk constructs, NA or M2e. The nanoparticles themselves can be quite immunogenetic, however some formulations do contain the adjuvant Matrix-M [24,47,83,84].

Recombinant influenza viruses expressing protein constructs developed for universal vaccine platforms have also been created using reverse genetics. This system allows for the production of live virus that express these proteins instead of traditional viral proteins. The recombinant viruses can replicate similarly to wild-type viruses and can therefore be used as seed strains in both IIVs and LAIVs that contain conserved epitopes not targeted by seasonal IIVs and LAIVs, however they still utilize the existing vaccine production pipeline [28,42,60,69,85,86]. As of 2020, 74 candidates have reached late pre-clinical development and 22 have reached clinical development with studies undergoing Phase 1–3 clinical trials (see detailed review [44]). This is where preclinical animal models are key. Animal models may indicate whether a vaccine candidate will be protective, and if so, correlates of protection can be determined by evaluating host immune responses in a variety of ways. Each animal model has advantages and disadvantages, which will be discussed in the next section. Choosing an applicable animal model is one of the most critical steps for vaccine studies.

## 4. Animal Models Used in Influenza Virus Vaccine Studies

Animal models have been extensively utilized to study influenza virus pathogenesis and transmission [41,82,83,84,85,86,87,88,89,90,91,92,93,94,95,96,97,98,99,100,101,102,103]. They were instrumental in the early isolation of the virus as well as its subsequent propagation. In the late 1930s and early 1940s, scientists began to look at virus pathogenesis in both mice and ferrets [104,105].

While vaccine safety is no longer evaluated for annually reformulated seasonal vaccines, animal models are still used in studies that focus on improving immunogenicity and efficacy. This usually involves testing whether adding novel adjuvants, increasing antigen dosing or using prime-boost methods can induce better immune responses compared to a standard vaccine regimen [93,106,107,108,109]. Novel vaccines must show efficacy, immunogenicity and safety in animal models before receiving approval for use in humans. This is especially important for evaluating vaccines focused on viruses with pandemic potential because protection studies cannot be ethically performed in human clinical trials [88]. For most vaccines, a tier of animal models is used to evaluate immunogenicity and efficacy of vaccines. This tier generally moves from smaller and more manageable models, such as mice, to larger ones such as ferrets or non-human primates. There are many factors involved in choosing the correct animal model for any vaccine study. Understanding the utilities of each animal model can aid in developing vaccine candidates that will perform well in clinical trials. The presence of clinical signs, data that can be collected, costs and husbandry requirements should be considered before choosing an animal model for a particular study. Additionally, finding animal models with sialic acid homology to humans is important in understanding not only viral pathogenesis, but also vaccine efficacy. The HA is responsible for host specificity and will bind preferentially to either α2,3 or α2,6 sialic acids depending on the subtype. Influenza viruses that infect humans have high affinity for α2,6 sialic acids, while those that infect avian species preferentially bind to α2,3 sialic acids. Sialic acids of the respiratory tract in ferrets and swine are predominately α2,6 linked, which has homology to humans [34,110,111]. Mice have a mix of both α2,3 and α2,6 sialic acids, with the former being more abundant [112,113]. The upper respiratory tract of guinea pigs is primarily composed of α2,6 sialic acids, however the lower respiratory tract has more α2,3 sialic acids [103]. Additionally, animal models can be manipulated to mimic some human comorbidities that are associated with increased risk of severe influenza virus infections or poor vaccine responses [94].

Animal models have been critical in establishing correlates of protection. Both vaccination and natural infection lead to a strong innate and adaptive immune response including the induction of Type 1 interferons, T cell activation, B cell maturation and antibody production [110,114,115,116]. Since vaccines are focused on inducing anti-HA antibodies, correlates of protection have primarily been determined by measuring the production of serum HA-specific antibodies that prevent binding to sialic acid receptors [48,49,50]. This is usually measured using a hemagglutinin inhibition assay (HAI). An HAI titer of >1:40 is considered as a surrogate for a protective antibody response. Interestingly, immune responses post-vaccination can vary compared to those induced by natural infection in regard to which virus proteins (i.e., the HA, NA, NP etc.) are primarily targeted by antibodies [32,33,51,117]. T-cells are also involved in clearing an infection, however traditional vaccines are not designed to induce strong T cell responses, while natural infection is more prone to inducing T-cell responses [111]. Novel influenza virus vaccines that target either the HA stalk, NA, M1, M2e or NP induce antibodies that cannot be detected in a hemagglutinin inhibition assay. With the development in new tools, it has been possible to understand other immune correlates of protection. By comparing antibody responses towards the NA after vaccination and natural infection, Chen et al. found that IIVs do not induce detectable anti-NA antibody titers while natural infection induces immune responses that are more balanced between the HA and NA [51]. This, and work from others, influenced the discovery of different correlates of protection, including anti-HA stalk antibodies and anti-NA antibodies [52,53,54]. As novel, T-cell based vaccines emerge, there will be more of an opportunity to examine T-cell driven correlates of protection. Information gained in animal models can be used to define primary outcome measures during clinical trials.

The main animal models involved in pre-clinical vaccine research are mice and ferrets, but other models such as the guinea pig, cotton rat, Syrian hamster, swine and non-human primates are also used (Figure 1). However, these models are uncommon and reagents for many downstream analyses are limited.

### 4.1. Mice

The mouse model is a staple in influenza virus vaccine and immunity studies. Generally, it is the first model used to investigate vaccine immunogenicity and efficacy. Vaccine studies using mice (*Mus musculus*) began by investigating whether delivering an inactivated form of influenza virus could serve as a surrogate to natural infection [118]. After the confirmation that the vaccine was immunogenic and safe in mice, the study continued and led to the development of the first inactivated influenza virus vaccine [19,119]. Almost all vaccines on the market, as well as those still in development, have been initially evaluated in mice.

While they are not a natural host for influenza viruses, mice can be experimentally infected with a variety of human influenza viruses [88]. However, pathology can vary widely between strains. Early isolates, such as A/Puerto Rico/8/1934 (H1N1) and B/Lee/1940 (IBV) are still used today as representative strains due to their high infectivity in mice. However, these viruses were adapted before their use in the mouse model. A/Puerto Rico/8/1934 was passaged 77 times in mice, 717 times in cell culture, 80 times in embryonated chicken eggs and 5 times in ferrets before it was used as the first inactivated vaccine strain in 1938. B/Lee/1940 was also passaged through embryonated chicken eggs before its use in the first bivalent vaccine. Both strains have been continuously passaged through eggs over time, making them more adept at causing severe infections in mice [117]

Additionally, many pandemic strains such as the H1N1 2009, H3N2 Hong Kong 1968, H5N1 and H7N9 viruses have been found to be lethal in the mouse model. Recent strains of human H1N1, H3N2 and IBV viruses are either unable to infect or need to be adapted to the mouse model [93,120]. For vaccine studies, a challenge virus that has been recently circulating in the population would be valuable as it would more accurately represent what humans would experience during clinical trials.

Mouse adapting is a key tool used to make challenge virus strains that can easily infect mice and cause clinical signs [121]. This process involves infecting mice with the wild type strain, collecting mouse lungs at one to three days post infection, homogenizing lung tissues and then infecting another mouse with diluted lung homogenates. Some viruses are easily adapted to mice, however other viruses may take several passages before they are fully adapted [122,123,124,125]. Mouse adaptation can cause mutations in many viral genes, including the polymerase machinery (PB1, PB2 and PA) as well as the glycoproteins HA and NA [126,127,128]. However, there has been little evidence of antigenic drift between the parental and mouse adapted viruses. Typically, wild type mouse strains such as BALB/c or C57BL6 mice are used for mouse adaptation. However, more susceptible mice, such as the DBA/2 strain or pharmacologically induced immune-suppressed, have been used to adapt viruses that are unable to infect wild type mice [126].

Mice are used to establish vaccine regimens (single dose, prime-boost and/or the addition of adjuvants), test routes of administration (IN, IM or intradermal), determine protective doses and immunogenicity as well as measure the reduction of morbidity, mortality and viral loads [88,91]. Mice are also often used to evaluate how vaccines can protect against homologous, heterologous and pandemic viral challenge. Vaccine studies are generally set up by immunizing groups of mice with differing regimens. Control groups, such as a standard of care (TIV/QIV), unvaccinated or sublethally infected mice are critical for understanding the differences in immune responses, protection and morbidity are induced by the vaccine and not by other factors. Experiments are designed to follow the vaccine distribution protocol and determine how the vaccine protects mice from lethal virus challenge.

Mice generally begin to show seroconversion approximately three weeks after vaccination, which is similar to the kinetics of seroconversion in humans. Once infected, unvaccinated mice generate high viral titers in their lungs at three days post infection that decrease by six days post infection. Clinical signs for infection in mice include lethargy, anorexia, weight loss and occasionally paralysis. Weight loss, as well as clinical signs, are used to measure morbidity prevented by the vaccine. Survival after viral challenge is recorded to determine the protectiveness of the vaccine [87,93,96,126]. Experimental setups are simple, with mice grouped based on their treatment and co-housed with 5–10 mice per cage, depending on the institution (Figure 2a). To study immune responses after vaccination tissues such as the lungs, trachea, lymph nodes, bone marrow and spleen can be collected. Sera is also collected to evaluate humoral immune responses. Bronchoalveolar lavage fluid (BALF) can be collected to determine viral titers and measure humoral immune responses. Tissues may be used for histology, determining viral titers or investigating T cell and B cell responses. Histology can visualize local inflammation, viral replication and immune infiltration into infected tissues. Determining viral titers gives a glimpse into how well the vaccine is preventing initial infection and further viral replication. By evaluating humoral responses, T cell activation and B cell maturation, we can determine how well the vaccine induces an adaptive immune response. Sera can be collected longitudinally and used to determine longevity of humoral responses. Generally, sera are assessed for binding to viral proteins via an enzyme linked immunosorbent assay (ELISA). Mice will also be evaluated for seroconversion using a hemagglutinin inhibition assay (HAI) and microneutralization assay (MN). These three assays are the staple to pre-clinical vaccine development in the mouse model [129].

There are few limits to reagents that are available to study host responses to vaccination in the mouse model, which is one of its biggest advantages [120]. In influenza virus research, the wild type strains C57BL/6 and BALB/c are generally used to mimic healthy individuals [88]. These mice have innate and adaptive immune responses like what would be seen in humans. However, some discrepancies in antibody subtypes and Th1/Th2 responses can be observed [130]. Humanized mouse models have emerged as a possible way to better utilize the mouse model for influenza virus vaccine research [131]. In one study, researchers designed a humanized mouse model to identify vaccine constructs that may cause adverse reactions in humans [132]. Another study developed a humanized mouse model to better study T cell mediated immunity by engrafting CD34+ hematopoietic progenitor cells into severely immunocompromised mice. The T cell repertoire after vaccination closely resembled what is observed in humans making this model viable for evaluating T cell-based vaccines [133].

It has been shown that mice upregulate similar genes as humans in response to infection [96]. DBA/2 mice are more susceptible to infection and may be used to study viruses that are generally not infectious in wild type mice [134,135]. Mouse models are inbred, with little genetic variation between mice of the same strain, which can have some benefits in vaccine studies such as the increased reliability that results obtained in one laboratory can be comparable to results reported elsewhere.

### 4.2. Ferrets

Ferrets (*Mustela putorius furo*) have been used in influenza virus research since the virus was first identified [2,3,20,109,112]. Ferrets are the most biologically relevant small animal model to study influenza virus pathogenesis, transmission and vaccinology [83,84,85,86,99]. There are many factors that make ferrets a more biologically relevant model including their ability to be naturally infected with human isolates, their sialic acid distribution homology with humans, their ability to transmit viruses through the air and their ability to mount protective immune responses after vaccination. They are also used extensively for risk assessment of viruses with pandemic potential to determine how infectious and transmissible a particular isolate is [89,136,137].

Like humans, the ferret respiratory track is lined with α2,6 sialic acids [34,138]. When infected with seasonal viruses, ferrets display clinical signs similar to humans including nasal discharge, anorexia, fever, sneezing and coughing [84,99]. Infection with zoonotic viruses can cause severe weight loss, pneumonia and a prolonged high fever. Additionally, ferrets are a unique model because they can transmit influenza viruses through direct contact and aerosols [83,84,85,86]. Clinical signs, along with quantifying viral loads in nasal secretions and body tissues, are used to determine how well a given vaccine protects from infection. Because ferrets’ signs and symptoms are like those in humans, these animals provide better clarity as to how the vaccine will behave in humans.

Experimental setups can vary depending on what questions are being asked about a particular vaccine [56,92,106,109,136,137,139,140,141,142,143,144,145,146,147,148,149,150,151,152,153,154,155] (Figure 2b). Ferrets are commonly used to assess how well a vaccine protects from morbidity and transmission. Additionally, host antibody responses can be evaluated to determine whether a vaccine induced protective HAI titers [113,116,156]. Ferrets are typically infected via direct administration of virus into each nostril [85]. For transmission experiments, ferrets will be separated into three groups, the donor (directly infected), contact recipient (uninfected) and aerosol recipient (uninfected). The vaccine can be administered to the donor ferret to assess whether vaccination prevents initial infection and further transmission to a naïve ferret. Alternatively, recipient ferrets can receive the vaccination to assess how well the vaccine protects from transmissible influenza virus. Contact recipients are co-housed with donor ferrets so that transmission occurs through frequent interactions and shared space. Influenza viruses can readily transmit through the direct contact route. Aerosol transmission is modeled by placing ferrets in separate cages with enough space between them to prevent physical contact. Airflow is designed to travel from the donor to the recipient ferret to aid in transmission, which can vary in effectiveness depending on the virus strain. For efficient transmission, ferrets should be housed with controlled temperature and humidity settings (ideally 23 °C with 30% relative-humidity), however, these practices are not yet standardized [157]. To closely mimic human vaccination strategies, ferrets are generally vaccinated following the same protocol that will be used in future clinical trials. Ferrets are infected intranasally by either using a pipet to drip virus into their nasal cavity or via aerosols, however the former is more efficient despite it being an unnatural route of infection [84,85].

In addition to monitoring clinical signs, many different specimens can be collected from ferret challenge studies allowing for an in-depth examination of vaccine efficacy. Prior to viral challenge, blood is collected from vaccinated animals to determine pre-infection antibody titers. Nasal washes or swabs can be collected throughout infection to determine viral titers, investigate innate immune responses in the form of cytokine and chemokine expression, and measure mucosal antibody titers [84]. The larger size of ferrets allows for the collection of tissues from many different anatomic compartments such as lobes of the lung, the trachea, and the nasal cavity allowing for the determination of viral spread within the animal [89,90]. This gives insight into how well a vaccine protects from both initial infection and virus spread by measuring viral titers as well as inflammation and infiltration of immune cells into infected tissues (histopathology). Collection of sera and BALF allows for the quantification of immune responses including innate immune responses (cytokine and chemokine expression), antibody levels towards viral proteins and vaccine antigens as well as the induction of cellular responses (for a detailed review see [84]). Previous vaccination experiments have confirmed that ferrets can mount strong, broadly reactive and relatively long-lasting immune responses to universal vaccine candidates. With promising results in the mouse and ferret models, vaccine candidates can be further tested in clinical trials [143,145,146,149,150,151,155]. In addition, because of their long life-span, it is possible to reuse ferrets from previous vaccine or pathogenesis studies to further examine how immune memory influences vaccine responses [158]. Ferrets are larger animals than mice, so the increased requirements for husbandry and handling of these animals must be considered. Additional ethical and technical restrictions limit the number of animals that can be included in a particular study, reducing the amount of treatment groups that can be tested.

There are less tools developed for studying ferret immune responses to vaccination compared to those for the mouse model. Much of the previous information gleaned about ferret immune responses was done using reagents made for canine research [159]. However, significant progress has been made in recent years to increase the availability of species specific reagents for more accurate analyses of ferret immune responses [82]. After the full ferret genome was sequenced, technologies to study genes contributing to host immune responses were developed [160]. Using the sequenced genome, scientists were able to express recombinant, full-length ferret proteins, such as IFN-γ [161,162,163]. Using hybridoma technologies, monoclonal antibodies that target key proteins in ferret immune responses, such as immunoglobins, interferons, cytokines and chemokines, have now been produced and are available commercially [87,164,165]. Additionally, probes for flow cytometry and ELISpot assays have been developed [166]. These tools are important for evaluating the mechanism of protection given by the vaccine and for determining any possible correlates of protection. Aside from the increased training and husbandry costs required for ferrets, these animals are a good model for influenza virus vaccine studies [82].

### 4.3. Guinea Pigs

Guinea pigs (*Cavia porcellus*) are primarily used to study influenza virus transmission [102,167,168]. They do not exhibit any clinical signs when infected with influenza virus making them poor models for assessing how a vaccine prevents morbidity. However, virus can easily be detected in nasal washes and guinea pigs can transmit most influenza viruses through direct contact or aerosol transmission [72,102,168,169]. Transmission studies for guinea pigs are set up just like ferrets, with an infected donor guinea pig being either co-housed (contact transmission) or separated (aerosol transmission) with uninfected guinea pigs (Figure 2b). It has been well established that cooler temperatures (ranging from 4 °C to 23 °C) and 20–30% relative humidity are optimal [97]. Typically, nasal washes are performed every other day after infection, up to 10 days post infection. Sera is also collected prior to vaccination and at various timepoints during the study. It can be used to verify seroconversion and to measure the levels of vaccine specific antibody titers [156]. One study utilized guinea pigs to determine that intranasal vaccination using recombinant NA protein was sufficient to prevent transmission from an infected to naïve animal [72]. Additionally, tissues from the respiratory tract can be collect for histology staining or quantifying virus titers [97]. Guinea pigs do have an advantage of being inbred, meaning that there is little variation between animals regarding gene expression and immune responses. Fewer reagents are available to interrogate guinea pig immune responses. A dearth of reagents, along with no clinical manifestations of disease make this model unrealistic for a well-rounded evaluation of vaccine induced immunity.

### 4.4. Cotton Rats

Cotton rats (*Sigmodon hispidus* or *Sigmodon fulviventer*) are not commonly used to study influenza virus pathogenesis or vaccinology. Unlike mice, cotton rats are susceptible to a variety of Influenza A and B viruses and can be infected with natural isolates without prior adaptation [94,95,105]. Upon infection, cotton rats exhibit increases respiratory rates, weight loss and hypothermia. Clinical signs, along with quantifying virus titers in various tissues, are used to evaluate candidate vaccines. While cotton rats are a slightly better model compared to mice in regard to susceptibility to infection, they also cannot transmit viruses between each other [94,95,105]. This, coupled with a lack of reagents to examine immune responses, has limited the number of vaccine studies conducted with cotton rats. However, in recent years, many new reagents have been developed to evaluate immune responses to infection and vaccination [89]. Vaccine studies in cotton rats have been primarily focused on increasing seasonal vaccine efficacy. One study used cotton rats to determine that a single dose of a whole inactivated influenza virus vaccine induces lower levels of serum antibodies compared to a prime-boost method. However, the quality of immune response was blunted so that animals in both groups exhibited similar morbidity despite differences in antibody titers [170]. Another study evaluated how well a seasonal TIV (2006–2007 season) could protect from drifted viral strains, which provided valuable information regarding strain selection for seasonal vaccine reformulations [107]. Few studies have examined universal influenza virus vaccines; however, it has been shown that cotton rats mount a cross-protective immune response when infected with heterosubtypic virus strains [171]. When combining data from other animal studies, cotton rats can be a valuable addition to evaluating candidate vaccines.

### 4.5. Hamsters

The Syrian hamster (*Mesocricetus auratus*) model has been infrequently used to evaluate influenza virus vaccines. Studies in the 1970s–1980s used hamsters to characterize cold-adapted LAIVs and whole-virus IIVs [164,172,173,174,175,176,177]. In humans, the upper respiratory tract is approximately 33 °C while the lower respiratory tract is 37 °C. Therefore, cold-adapted viruses would be restricted to only the nasal cavity and could not cause lower respiratory tract infections like wild-type viruses. Hamsters were considered a good model because their upper and lower respiratory tracts were also 33 °C and 37 °C, respectively [165]. These animals also share sialic acid homology with humans [178,179,180,181]. In addition to studying LAIVs, hamsters were used extensively to understand heterosubtypic immunity and the implications of immune history (either from vaccination or natural infection) on vaccine responses [182,183,184].

To evaluate vaccine responses, sera, nasal washes and respiratory tissues can be collected [185]. Efficacy was measured by the ability of a vaccine to induce serum hemagglutinin inhibition titers along with the reduction of virus titers in nasal washes and respiratory tissues [164,172,173,174,175,176,177]. Now, Syrian hamsters are an uncommon animal model for influenza virus research, especially for vaccine studies.

Hamsters have several advantages including their natural susceptibility to most human influenza viruses and their ability to transmit via direct contact or aerosol routes [164,185]. These animals are also relatively small and easy to maintain, making them an attractive animal model. However, they do not exhibit clinical signs, although some pdmH1N1 viruses cause mild weight loss at high doses [185]. There are also few reagents available to investigate immune responses to vaccination. In recent years, work has begun to develop genetically modified hamsters to mimic human diseases [167]. Immunological tools, such as antibodies targeting specific hamster immune response proteins are also being created [168]. Once improved tools are available, the hamster model could become an important small animal model for vaccine research. However, they share many similarities with other rodent models, such as guinea pigs or cotton rats, making their utility redundant for many studies.

### 4.6. Swine

After the outbreak of swine-origin pdmH1N1 in 2009, there became a renewed interest in evaluating domestic swine (*Sus scrofa domesticus*) as a model for influenza vaccine research [95,169]. There are more than 10 different H1 and H3 clusters of viruses that co-circulate in North American swine [101]. Additionally, there is frequent avian-to-swine and human-to-swine IAV transmission events, leading to significant viral diversity in the swine population [186,187]. If co-infected with human and swine IAVs, swine can act as a “mixing vessel” where there is a chance of reassortant strains emerging that carry both swine-origin and human-origin viral genes [178,182,183,188]. While these reassortant strains are rare, one emerged in 2009 that contained swine, avian and human-origin genes, leading to the 2009 H1N1 pandemic [184]. Like avian species, there is constant global surveillance of viruses circulating in swine in an attempt to curb the emergence of future viruses with pandemic potential [183]. Swine can transmit influenza viruses to each other, and humans, through direct contact and aerosols. Swine present mild to moderate symptoms when infected including nasal discharge, labored breathing, fever, coughing and weight loss [100,106,189]. Infections are rarely fatal, however swine influenza virus vaccines are available to prevent outbreaks that could have an economic impact on the swine industry [179]. Like in humans, swine mount cross-reactive antibody responses when immunized, however unlike humans, their responses are not as cross-protective [180,190]. Studies have suggested that this phenomenon is caused by vaccine-associated enhanced respiratory disease (VAERD), where vaccinated swine exhibit worse morbidity compared to unvaccinated counterparts. VAERD has been primarily associated with heterologous viral challenge after receiving inactivated whole influenza virus vaccines [191]. While not yet observed in humans or other animal models, swine remain a good model and are increasingly used to evaluate vaccines [120].

### 4.7. Non-Human Primates

Non-human primates are not a natural host for influenza viruses, however they can be experimentally infected with human isolates [87]. Many different species have been used to study influenza virus pathogenesis and immunology including African green monkeys (*Chlorocebus sabaeus*), cynomolgus macaques (*Macaca fascicularis*), pigtail macaques (*Macaca nemestrina*), rhesus macaques (*Macaca mulatta*) and common marmosets (*Callithrix jacchus*) [181,192,193,194,195,196]. As primates, this animal model is the most like humans and can be used to determine the future success of vaccine candidates in human clinical trials. However, the use of non-human primates is limited because of the expensive husbandry costs and specialized facilities needed to house the animals. In addition, experimental sample sizes are kept to a minimum because of ethical concerns. These limitations are why non-human primates are not a staple in influenza virus vaccine research. They have been used to evaluate pandemic influenza virus vaccines, since those cannot be tested in humans [197,198,199]. Universal influenza virus vaccines have also been evaluated in non-human primates for similar reasons. While the protective efficacy of universal influenza virus vaccines can be determined in humans for seasonal viruses, the non-human primate model can evaluate both seasonal and pandemic influenza viral challenge.

Non-human primates exhibit similar signs and symptoms as humans when productively infected with influenza virus. Viral loads can be detected in nasal washes and in respiratory tissues after infection. Studies have found that non-human primates are able to mount cross-reactive and protective immune responses like what is seen in other animal models and human clinical data. In one study, a single DNA vaccination in combination with a seasonal vaccine booster was able to induce broadly reactive antibodies that protected from drifted H1N1 viruses [200]. Interestingly, not all non-human primates can transmit influenza viruses through the air. It was shown that the common marmoset can transmit viruses to each other, while rhesus macaques do not [194,201]. Without transmission events, the non-human primate model does not truly recapitulate the human population making other models, such as the ferret model, more applicable. Almost all candidates that have been evaluated using the non-human primate model have first been assessed in either a mouse or ferret model to ensure vaccine quality and determine effective dosages. Several universal influenza virus vaccine candidates have been evaluated in the non-human primate model [57,102,189,192,200,202,203,204].

## 5. Pre-Clinical Animal Models of High-Risk Populations

After the H1N1 2009 pandemic, it was found that pregnant and obese individuals had higher rates of hospitalization and death compared to the general population [205,206,207,208]. It has also been observed that malnourished, diabetic, young children and elderly individuals exhibit increased morbidity and mortality from influenza virus infections. Conversely, these populations also have reduced vaccine responses compared to healthy adults [197,209,210,211,212]. To better meet the needs of these vulnerable populations, vaccines need to be optimized by either increasing antigen doses, adding adjuvants or implementing prime-boost strategies. The field of modeling conditions related to decreased vaccine responses in animals is still being developed, however early vaccine studies in high-risk hosts have been conducted using mice, ferrets and non-human primates.

### 5.1. Mice

Mice are the easiest small animal model to use when mimicking human genetic or lifestyle related high-risk conditions. They can be easily manipulated to generate gene-specific knockout strains. For example, mice have been used to model genetic obesity by having either their leptin receptor (db/db) or leptin (ob/ob) genes knocked down [98,198]. C3−/− mice are compliment deficient and can be used to look at the contribution of antibody dependent cellular toxicity pathways to protection from infection [199]. This is important because many universal influenza virus vaccines induce antibodies that are non-neutralizing and rely on other cell-mediated immune mechanisms to provide protection [59,144]. Vitamin deficient mouse models have been used to evaluate immune responses in malnourished hosts [213,214]. One study reported that vitamin supplements given at the time of vaccination improved vaccine responses in deficient mice [215]. Mice can also be used to examine other human conditions such as infancy [216], malnutrition [215,217,218,219,220,221], aging [197,222,223,224], diet-induced obesity [103,219,225,226] or pregnancy [205,212,227,228,229,230,231,232,233]. The variety of populations that the mouse model can mimic allows for vaccines to be evaluated for multitude of individuals. Increased pathogenesis was recapitulated in obese and pregnant mouse models, showing that immune-compromised hosts mount suboptimal immune responses while having increased pathology after infection [208]. This leads to more severe disease and poor immune memory, leaving the host suspectable to more infections. Pregnant mice have been used to study maternal antibody transfer of antibodies from TIV vaccinated dams to their pups [232]. Aging mice have been used extensively to understand immune responses in elderly populations. Findings from these experiments include the fact that T cell function declines with age, reducing the effectiveness of vaccines [224]. Additionally, B cell function is dysregulated in aging mice, leading to lowered antibody responses and retardation of immune memory [234]. These observations led to new recommendations of high-dose vaccines for individuals ≥65 years of age [197,235,236,237]. Using these various mouse models, scientists have been able to not only determine immune pathways that are important during infection, but also determine how to exploit these pathways to generate stronger immune responses after vaccination.

### 5.2. Ferrets

Ferrets are a more complex model compared to mice, limiting the amount of characterized high-risk models. In addition, while a few transgenic ferret models have been developed [238,239], it is difficult to make specific genetic knockout strains to be as readily available as those for the mouse model. However, human conditions not influenced exclusively by genetics, such as immunocompromised states [142,240], diet-induced obesity [98], aging [241] and pregnancy [225,242] can be modeled in ferrets. Just like humans, ferrets that have comorbidities exhibit higher morbidity and mortality when infected with seasonal influenza viruses [99,103,220,221,226,243,244,245]. Chemotherapy treated ferrets have been used to characterize the safety and immunogenicity of an LAIV vaccine in an immunocompromised model [142]. Additionally, aged ferrets were used to study impaired immune responses after sequential infections with H1N1 viruses [241]. Novel vaccines are not frequently assessed in high-risk ferret models as of now, but these studies are expected to increase once vaccine candidates prepare for clinical trials.

### 5.3. Non-Human Primates

The non-human primate model for high-risk hosts is relatively underdeveloped. Few models have been characterized and utilized in influenza virus vaccine research. Nonetheless, vaccine research in neonatal and aged non-human primate models has helped in identifying the relative efficacy of a given vaccine for the population. Elderly rhesus macaques were used to examine whether the addition of an adjuvant to a traditional TIV vaccine improved immunogenicity [226]. Another elderly non-human primate model using cynomolgus macaques was used to characterize the pathogenesis of H7N9, a virus with pandemic potential [243]. Neonate African green monkey models have been used to characterize immune responses to novel adjuvants and natural infection [244,246,247]. Additionally, neonatal and adult African green monkeys have been used to characterize humoral responses after natural infection with a seasonal influenza virus, A/Puerto Rico/8/1934 [248]. While an obese non-human primate model is being developed, [249] there has not yet been research into how influenza vaccines behave in this model. Continued work in high-risk non-human primate models will increase the reliability of results found in both mouse and ferret models, allowing for more confident predictions on how vaccines will behave in high-risk humans.

## 6. Conclusions

When designing vaccination studies, scientists must decide on which animal model will best address their hypotheses for a particular stage of vaccine development. Many Institutional Animal Care and Use Committees (IACUC) encourage researchers to limit animals used and, when possible, try other methods to address their hypotheses. To that end, unconventional models are being developed for future influenza virus vaccine studies. One study used primary human nasal epithelial cells to evaluate the differences in innate immune responses after infection with wild type H3N2 or a matched LAIV [250]. By identifying key innate immunity pathways induced after exposure to influenza virus vaccine strains, a connection can be made to pathways that will lead to protective adapted immune responses. Further development of primary cell technologies may lead to a reduction in animals used in influenza virus research. It is also possible to conduct initial risk assessments using human and ferret primary cells isolated from different respiratory tract tissues to determine if there is any susceptibility to avian isolates [251,252,253,254,255,256]. If infection occurs, then ferrets would be used to examine transmission potential. However, ex vivo systems cannot replace the complexity required for a protective immune response to vaccination. While animal models have limitations, they remain invaluable for preclinical vaccine development.

Advances in the safety and immunogenicity of universal influenza virus vaccine candidates has reinvigorated the human challenge model [92,245,257,258,259,260,261,262,263,264,265,266,267,268,269,270]. Human challenge studies are conducted by immunizing participants with vaccine candidates and monitoring for any side effects or adverse reactions. After a minimum of four weeks, participants will be experimentally infected with influenza virus intranasally. Signs, symptoms and nasal washes and are collected for two weeks following challenge to evaluate efficacy of the vaccine. Participants can also be monitored longitudinally with sera collected every few months to determine long term immunity induced by the vaccine [262]. If vaccine formulations have proved to be safe and immunogenic in preliminary experiments, they could hypothetically move into the human challenge model instead of going through the tier of animal models.

Animal models have been the foundation for many of the new vaccine approaches in clinical trials today. When developing new vaccine technologies and evaluating correlates of protection, it is important to understand the benefits and limitations of each animal model. The husbandry abilities of animal facilities at research institutions, availability of reagents and other factors need to be considered. If a vaccine candidate is in early developments, it may be best to use the mouse or cotton rat for dosing, immunogenicity and efficacy of the vaccine. As the product is more polished, it will be important to move into a more biologically relevant model such as the ferret or non-human primates.

## Figures and Tables

**Figure 1 vaccines-09-00787-f001:**
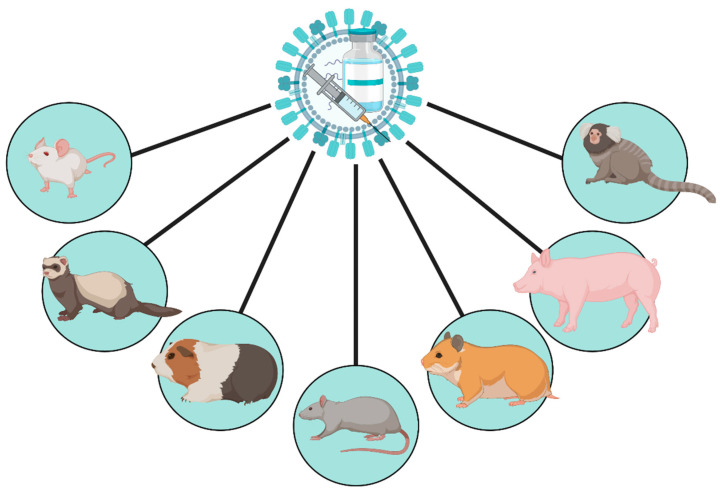
Animal models for influenza virus vaccine development. Mice, ferrets, guinea pigs, cotton rats, swine and non-human primates are important animal models for vaccine development. This figure was created with BioRender.com (accessed on 1 April 2021).

**Figure 2 vaccines-09-00787-f002:**
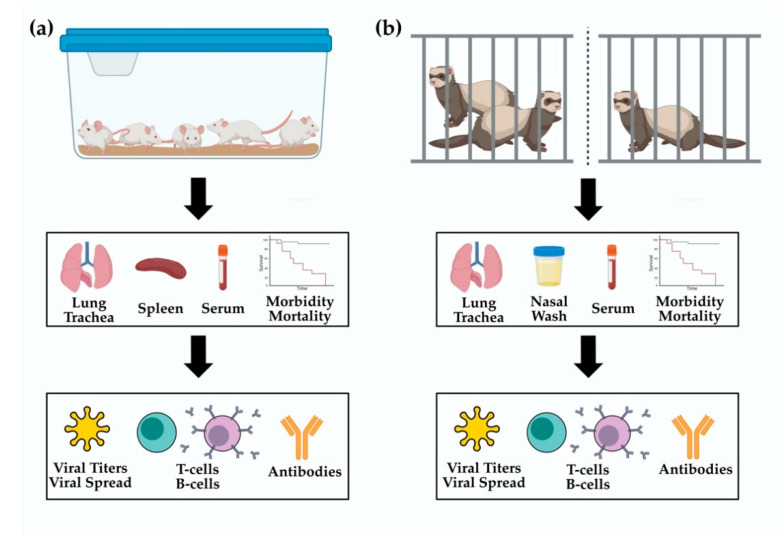
Experimental setups for vaccination experiments: (**a**) mouse vaccination experiments are designed so that each vaccine group is co-housed in a single cage. A maximum of five to ten mice can be placed in a single cage. Morbidity and mortality are monitored for 14 days post challenge. Sera, lungs, trachea and spleens may be collected to measure viral titers, viral spread, B-cell, T-cell and antibody responses; (**b**) ferret transmission and vaccination experimental design. For contact transmission, ferrets are co-housed with a maximum of two animals per cage. For aerosol transmission, ferrets are in two separate cages, spaced far apart enough to prevent direct contact. Airflow goes from the infected to uninfected animal. Tissues and sera are collected and utilized in a similar manner as those collected for mouse experiments. Additionally, ferret nasal wash samples or nasal swabs are collected and can be used to interrogate viral titers, innate immune responses and mucosal immunity. This figure was created with BioRender.com (accessed on 1 April 2021) and Affinity Designer v1.9.

**Table 1 vaccines-09-00787-t001:** Influenza virus vaccines currently on the US market [27].

Trade Name	Manufacturer	Category	Available as	Demographic
AFLURIA	Seqirus Pty. Ltd.	IIV	TIV, QIV	Persons > 6 months of age
Agriflu	Seqirus Inc.	IIV	TIV	Persons > 18 years of age
FLUAD ^a^	Seqirus, Inc.	IIV	TIV	Persons > 65 years of age
FluMist	MedImmune	LAIV	QIV	Persons 2–49 years of age
Fluarix	GlaxoSmithKline Biologicals	IIV	TIV	Persons > 3 years of age
Fluarix Quadrivalent	GlaxoSmithKline Biologicals	IIV	QIV	Persons > 6 months of age
Flublok	Protein Sciences Corporation	Recombinant HA	TIV, QIV	Persons > 18 years of age
Flucelvax	Seqirus, Inc.	IIV	TIV	Persons > 4 years of age
Flucelvax ^b^	Seqirus, Inc.	IIV	QIV	Persons > 4 years of age
FluLaval	ID Biomedical Corporation of Quebec	IIV	TIV, QIV	Persons > 6 months of age
Fluvirin	Seqirus Vaccines Limited	IIV	TIV	Persons > 4 years of age
Fluzone ^c^	Sanofi Pasteur Inc.	IIV	TIV, QIV	Persons > 6 months of age
Influenza Virus Vaccine, H5N1 ^d^	Sanofi Pasteur Inc	IIV	Monovalent	Persons 18 through 64 years of age
Influenza A (H5N1) Virus Monovalent Vaccine ^e^	ID Biomedical Corporation of Quebec	IIV	Monovalent	Persons > 6 months of age

^a^ Adjuvanted with MF59^®^; ^b^ cell-based vaccine; ^c^ available as a standard dose IM vaccination, high-dose IM vaccination and intradermal vaccination; ^d^ for national stockpile; ^e^ adjuvanted with AS03.

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
