# Peer review of "Animal Models Utilized for the Development of Influenza Virus Vaccines"

_vaccines, 2021, doi:10.3390/vaccines9070787_

Round 1

Reviewer 1 Report

It was a generally well-written review describing animal models typical used in the influenza virus vaccine research. The comments below would need to be addressed.

-The author could tune down details on how the experiment was performed. Instead, stressing the advantage and disadvantage of each animal model.

-Can the author write a little bit on the “dirty” mouse model, it’ll be of interest to the audience

-Animal models that are susceptible to toxicity challenge might be different from those to assess immunogenicity and efficacy. Such as rats, rabbits, ferrets, etc. So when the authors mention animal models for safety evaluation, it could be interpreted as animal models for toxicology study.

-Line 309, The statement that “These mice have innate and adaptive immune responses like what would 309

 be seen in humans.” Maybe a little discussion on humanized mouse model used for flu vaccine research?

-Line 128, ASO3 should be AS03

-line 172, through reverse genetics, the virus would be recombinant. It perhaps more accurate to say modified protein constructs

-Line 199, larger animals includes ferrets and NHP

-Line, it’s perhaps more accurate to say “Animal models used in influenza virus vaccine studies”

-Line 257, please add references for the statement “Pre-pandemic, seasonal H1N1 viruses are better suited for mouse infection”. The next sentence makes it sounds like both A/PR/8/1934 and B/Lee/1940 are both pre-pandemic seasonal H1N1, which was not the intention. Also, did the currently lab strain A/PR/8/1934 and B/lee/40 mouse-adapted at some point?

-For mouse adaptation, which mouse strains were typically used?

-Line 294, BALF is not considered a tissue

Author Response

  • The author could tune down details on how the experiment was performed. Instead, stressing the advantage and disadvantage of each animal model.
    • Thank you for this comment. We have toned down the experimental details.
  • Can the author write a little bit on the “dirty” mouse model, it’ll be of interest to the audience
    • We agree that this is a very interesting model. However, we were unable to find published vaccine (or pathogenesis) studies that use a dirty mouse model. Thus, this model was not discussed.
  • Animal models that are susceptible to toxicity challenge might be different from those to assess immunogenicity and efficacy. Such as rats, rabbits, ferrets, etc. So when the authors mention animal models for safety evaluation, it could be interpreted as animal models for toxicology study
    • We are referencing toxicology by using the word safety. If any animal model appeared to have adverse reactions to a vaccine it would not be considered safe for humans regardless of immunogenicity or efficacy. There were some places in the text where this was unclear and they have been changed for clarity.
  • Line 309, The statement that “These mice have innate and adaptive immune responses like what would be seen in humans.” Maybe a little discussion on humanized mouse model used for flu vaccine research?
  • Agreed. A description of a few studies using humanized mice has been added to the text.
  • Line 128, ASO3 should be AS03
    • This has been corrected in the text.
  • Line 172, through reverse genetics, the virus would be recombinant. It perhaps more accurate to say modified protein constructs
    • The text has been modified to clarify that the recombinant viruses (generated through reverse genetics) contain proteins from universal vaccine platforms.
  • Line 199, larger animals includes ferrets and NHP
    • The reference to NHP has been added to the text.
  • Line, it’s perhaps more accurate to say “Animal models used in influenza virus vaccine studies”
    • I believe this is referring to line 185. The change has been added to the section title.
  • Line 257, please add references for the statement “Pre-pandemic, seasonal H1N1 viruses are better suited for mouse infection”. The next sentence makes it sounds like both A/PR/8/1934 and B/Lee/1940 are both pre-pandemic seasonal H1N1, which was not the intention. Also, did the currently lab strain A/PR/8/1934 and B/lee/40 mouse-adapted at some point?
  • The sentence “Pre-pandemic, seasonal H1N1 viruses are better suited for mouse infections” has been removed for clarity. PR8 has been defined as H1N1 and B/Lee has been defined as an IBV. The passage history of PR8 was also added to the text. We have also elaborated that these viruses have had to be passaged frequently over the years and that this has increased their likelihood of being lethal in the mouse model.
  • For mouse adaptation, which mouse strains were typically used?
  • Typically, wild type mouse strains such as balb/C or C57BL6 mice are used for mouse adaptation. However, more susceptible mice, such as DBA2 strains or pharmacologically induced immune-suppressed, have been used to adapt viruses that are unable to infect wild type mice” has been added to the text to define which mouse strains are commonly used for mouse adaptation.
  • -Line 294, BALF is not considered a tissue
    • This has been corrected. An elaboration of what BALF can be used for is also added to the text.
    •  

Reviewer 2 Report

Ericka Kirkpatrick Roubidoux and Stacey Schultz-Cherry have submitted the review entitled “Animal models utilized for the development of influenza virus vaccines”. This is an excellent and timely needed review. The authors have very clearly discussed different types of models used in the influenza vaccine development, including the practical issues, and suggested suitable animal models for different types of experimental setups. The manuscript is well written and easy to follow.

I hereby endorse the manuscript for publication in its present form.

Author Response

Thank you

Reviewer 3 Report

The review by Roubidoux and Shultz-Cherry provides a nice oversight into the factors influencing the selection of different animal models for the pre-clinical assessment of experimental influenza vaccines. The text is largely clear, although some the phrasing perhaps a bit too conversational in places. The sections on each animal model might better spell out the strengths and weaknesses of each animal as they pertain specifically to vaccine testing. Overall a good addition to the literature, I have some minor comments below:

Line 76-77 – “LAIVs are administered based on an infectious dose, meaning that other viral proteins, such as the NA, are also included.”    - The authors should note here that IIV often also includes other proteins such as NA, just that the amounts are not controlled for.

Line 121-122 – “IIVs induce strong immune responses without the addition of an adjuvant“  - On what basis are they “strong”.  Could be said to be weak compared to say the yellow fever immunisation or COVID vaccines.  Suggest rephasing to make clear the immunity induced by unadjuvanted IIV is sufficient for a degree of seasonal protection.

Line 126-127 – “While they are not commonly used, there are a few adjuvants 126 that are licensed for uses in influenza virus vaccines.” – The authors might like to briefly clarify the circumstances under which adjuvanted vaccines are recommended and any data supporting that they are actually more efficacious.

Line 223 – 224 – “Interestingly, immune responses post-vaccination can vary compared to those induced by natural infection“   - Immunity to infection is also highly variable. Not sure what the authors are trying to say here, suggest clarifying

Line 285 – “Mice generate immune responses like humans…”  - this is worded strangely, suggest rephrasing to make clear you are talking about the kinetics here.

Line 292 – 294 – “To study immune responses after vaccination tissues such as the lungs, trachea, bronchialveolar lavage fluid (BALF) and spleen can be collected.”  - No mention here of the lymph nodes, which are presumably the site of generation of the humoral response.  Nor the bone marrow, where plasma cells reside. 

Line 306 – 307 – “There are no limits to reagents that are available to study host responses to vaccina- 306 tion in the mouse model.” – this might be overstating things a little, certainly the availability of some reagent differs between mice and say human experiments.

Line 314 – “which can have some benefits in vaccine studies.”  - Such as?

Line 327 – “While ferrets are not people..”  - ??

Line 332 – “ferrets’  the ferret

Line 339 – “Because ferrets’ signs and symptoms are like those in humans, these animals provide better clarity as to how the vaccine will behave in humans.”  - I think this point is contestable, while undoubtedly displaying more human-like lung physiology, there is limited evidence to suggest the ferret is closer to humans immunologically than other mammalian species.

Line 515 – 516 -  “Non-human primates exhibit similar signs and symptoms as humans when infected 515 with influenza virus.” – this statement is too broad, a lot of viruses are asymptomatic or fail to reliably establish in many primate models.

Line 343-344 – “Additionally, host immune responses can be evaluated to determine any correlates of protection…”  - As before, I would find it hard to agree that ferrets are a good model for correlates analysis other than gross antibody responses, given so little is know immunologically in this animal.  Great for looking at transmission and morbidity yes, but for immunology studies still a long way to go.

Line 398 – “ferrets, these animals are an ideal model for influenza virus vaccine studies“  - I think this point need better support.  Ideal in what sense, that they accurately predict vaccine efficacy in humans?  This has not been established.  I think the model needs a fairer balance of strengths and weaknesses, in particular spelling out what knowledge gaps need to be closed to have more confidence in the ferret as a good immunological model.

Author Response

  • Line 76-77 – “LAIVs are administered based on an infectious dose, meaning that other viral proteins, such as the NA, are also included.”    - The authors should note here that IIV often also includes other proteins such as NA, just that the amounts are not controlled for.
    • This clarification has been added to the text.
  • Line 121-122 – “IIVs induce strong immune responses without the addition of an adjuvant“  - On what basis are they “strong”.  Could be said to be weak compared to say the yellow fever immunisation or COVID vaccines.  Suggest rephasing to make clear the immunity induced by unadjuvanted IIV is sufficient for a degree of seasonal protection.
    • “Strong” was changed to “protective” in the text.
  • Line 126-127 – “While they are not commonly used, there are a few adjuvants that are licensed for uses in influenza virus vaccines.” – The authors might like to briefly clarify the circumstances under which adjuvanted vaccines are recommended and any data supporting that they are actually more efficacious.
    • A follow up sentence has been added. The use of adjuvants in vaccines is also discussed in the mouse animal model section, where multiple studies are referenced.
  • Line 223 – 224 – “Interestingly, immune responses post-vaccination can vary compared to those induced by natural infection“   - Immunity to infection is also highly variable. Not sure what the authors are trying to say here, suggest clarifying
    • This has been clarified to emphasize that vaccination induces antibodies towards the HA while natural infection induces antibodies to the HA, NA, NP etc.
  • Line 285 – “Mice generate immune responses like humans…”  - this is worded strangely, suggest rephrasing to make clear you are talking about the kinetics here.
    • This has been clarified.
  • Line 292 – 294 – “To study immune responses after vaccination tissues such as the lungs, trachea, bronchialveolar lavage fluid (BALF) and spleen can be collected.”  - No mention here of the lymph nodes, which are presumably the site of generation of the humoral response.  Nor the bone marrow, where plasma cells reside. 
    • Agreed. This is now stated in the text.
  • Line 306 – 307 – “There are no limits to reagents that are available to study host responses to vaccina- 306 tion in the mouse model.” – this might be overstating things a little, certainly the availability of some reagent differs between mice and say human experiments.
    • “No” has been replaced with “few”
  • Line 314 – “which can have some benefits in vaccine studies.”  - Such as?
    • We have added that a major benefit is that the results collected from different institutions can be more easily compared since the mice used in all experiments are isogenic.
  • Line 327 – “While ferrets are not people..”  - ??
    • This has been changed (see reviewer 4).
  • Line 332 – “ferrets’ à the ferret
    • This has been corrected.
  • Line 339 – “Because ferrets’ signs and symptoms are like those in humans, these animals provide better clarity as to how the vaccine will behave in humans.”  - I think this point is contestable, while undoubtedly displaying more human-like lung physiology, there is limited evidence to suggest the ferret is closer to humans immunologically than other mammalian species.
    • We agree that there is limited data on the relatedness of human and ferret immunology (which will become less limited as new reagents are produced/used). However, this sentence is only referring to determining virus titers, signs and symptoms which are commonly reported in pre-clinical ferret vaccine studies.
    •  
  • Line 343-344 – “Additionally, host immune responses can be evaluated to determine any correlates of protection…”  - As before, I would find it hard to agree that ferrets are a good model for correlates analysis other than gross antibody responses, given so little is know immunologically in this animal.  Great for looking at transmission and morbidity yes, but for immunology studies still a long way to go.
    • The text has been modified to refer to protective antibody titers instead of the broad statement of “immune responses”.
    •  
  • Line 398 – “ferrets, these animals are an ideal model for influenza virus vaccine studies“  - I think this point need better support.  Ideal in what sense, that they accurately predict vaccine efficacy in humans?  This has not been established.  I think the model needs a fairer balance of strengths and weaknesses, in particular spelling out what knowledge gaps need to be closed to have more confidence in the ferret as a good immunological model.
    • We agree and have modified the text accordingly.
  • Line 515 – 516 - “Non-human primates exhibit similar signs and symptoms as humans when infected with influenza virus.” – this statement is too broad, a lot of viruses are asymptomatic or fail to reliably establish in many primate models.

The word “productively” has been added in front of infected

Reviewer 4 Report

This is an excellent review and resource for the field.  I have only minor suggestions:

  1. minor edit on line 130 "moved into clinical trials"
  2. line 140, WHO was already defined
  3. Line 224 is an interesting statement, and more explanation would be helpful to the reader.
  4. For each animal model:
    1. What is the primary endpoint (main measure of disease or protection)?
    2. Are both males and females used, or is one gender preferred?
    3. What is the typical age/weight range used?
    4. What is the typical n= per group?
    5. Timing of disease progression, detection of virus and emergence of antibody response?
  5. Twice, it is mentioned that only 5 mice can be housed per cage, but at my institution, 10 mice can be housed per cage (perhaps we have bigger cages).
  6. For the Ferret model, line 323, may want to mention nasal swabs.  Also, with regard to washes vs swabs, is there an advantage of one versus another (consistency between data points and/ or yield)?
  7. Line 327, I recommend editing the sentence to "Ferrets are the most biologically relevant model...."
  8. Lines 327-328, could you list why Ferrets are the most biologically relevant animal model following this sentence?
  9. Line 354, a figure or photograph/diagram showing how transmission experiments/with directional airflow, are set up would be helpful.
  10. Line 338, minor edit, "given" instead of "give"
  11. Line 357, are ferrets typically inoculated in both nares for the IN model?  Is there a typical volume?
  12. Line 380, should the cooler room temperature requirement for Ferrets be mentioned?  At our facility, this is a challenge because not all rooms can be set to a specific temperature.
  13. Line 385, use "gleaned" instead of "gleamed"
  14. For both cotton rats (Line 428) and NHPs (line 520), it is mentioned that they cannot transmit influenza.  Would it be more appropriate to state that "transmission has not been demonstrated", since it is difficult to demonstrate a negative result, or possibly infrequent event?
  15. Line 450, are these temperatures similar to humans?
  16. Line 473, what year(s) did this outbreak take place?
  17. Lines 478-480, are these observations only from natural occurrences or experimentally demonstrated as well?
  18. Lines 517-518, more detail would be nice.
  19.  Line 570, what are the transgenic ferret models, specifically?
  20. Line 510 and Final paragraph:, are GLP/Animal Rule Studies in NHPs (and/or other species) used to get FDA approval of vaccines that are meant for stockpiling for possible future  pandemics?

Author Response

    • minor edit on line 130 "moved into clinical trials"
      • This has been corrected.
    • line 140, WHO was already defined
      • This has been corrected.
    • Line 224 is an interesting statement, and more explanation would be helpful to the reader.
      • The text has been modified for clarity (see response to reviewer 3)
    • For each animal model:
      • What is the primary endpoint (main measure of disease or protection)?
      • Are both males and females used, or is one gender preferred?
      • What is the typical age/weight range used?
      • What is the typical n= per group?
      • Timing of disease progression, detection of virus and emergence of antibody response?       
        • Primary endpoints have been described as an HAI titer >1:40 as well as a reduction of virus titers, transmission events and morbidity. Interestingly, the progression of immune responses and disease symptoms (where applicable) are similar to humans. Many of the other details are not standardized/reported and would be difficult to summarize.
    • Twice, it is mentioned that only 5 mice can be housed per cage, but at my institution, 10 mice can be housed per cage (perhaps we have bigger cages).
      • That’s interesting that your institution has 10 animals per cage. To clarify, we have changed it to “5-10 mice per cage, depending on the institution”.
    • For the Ferret model, line 323, may want to mention nasal swabs.  Also, with regard to washes vs swabs, is there an advantage of one versus another (consistency between data points and/ or yield)?
  • We added nasal swabs to the figure description. Nasal washes are generally better for determining virus titers because you “wash” the nasal passages versus just swabbing the nose.
  • Line 327, I recommend editing the sentence to "Ferrets are the most biologically relevant model...."
    • This has been changed.
  • Lines 327-328, could you list why Ferrets are the most biologically relevant animal model following this sentence?
    • A follow up sentence has been added.
  • Line 354, a figure or photograph/diagram showing how transmission experiments/with directional airflow, are set up would be helpful.
    • Arrows have been added to figure 2B to show the direction of airflow.
  • Line 338, minor edit, "given" instead of "give"
    • This has been corrected.
  • Line 357, are ferrets typically inoculated in both nares for the IN model?  Is there a typical volume?
    • Ferrets are inoculated in both nares and inoculation volumes range from 500uL to 1mL (Belser 2016). The route of administration has been added to the text.
  • Line 380, should the cooler room temperature requirement for Ferrets be mentioned?  At our facility, this is a challenge because not all rooms can be set to a specific temperature.
    • The ideal transmission conditions have been added to the text. Also, we added similar details for guinea pig transmission experiments.
  • Line 385, use "gleaned" instead of "gleamed"
    • This has been changed.
  • For both cotton rats (Line 428) and NHPs (line 520), it is mentioned that they cannot transmit influenza.  Would it be more appropriate to state that "transmission has not been demonstrated", since it is difficult to demonstrate a negative result, or possibly infrequent event?
    • The literature describing those models contains experiments where transmission experiments were attempted, and no transmission was observed. There may be rare transmission events but cotton rats and some NHPs would not be a reliable model for virus transmission.
  • Line 450, are these temperatures similar to humans?
    • Yes, it is described in the text.
  • Line 473, what year(s) did this outbreak take place?
    • The year (2009) has been added to the text.
  • Lines 478-480, are these observations only from natural occurrences or experimentally demonstrated as well?
    • The mixing vessel theory is supported by comparing sequences from viruses isolated from humans, swine, and avian species. The pdm-H1N1 virus contained genes from swine influenza viruses, human influenza viruses and avian influenza viruses. It has also been demonstrated in vitro but is difficult to recapitulate in vivo due to “gain of function” experiment restrictions.
  • Lines 517-518, more detail would be nice.
    • A follow up sentence has been added.
  •  Line 570, what are the transgenic ferret models, specifically?
    • Neither model referenced is used for influenza virus research. One is for modeling cystic fibrosis and another is a CRISPR/Cas9 system for mimicking developmental abnormalities.
  • Line 510 and Final paragraph:, are GLP/Animal Rule Studies in NHPs (and/or other species) used to get FDA approval of vaccines that are meant for stockpiling for possible future  pandemics?
    • There have not been any Animal Rule Studies regarding pandemic influenza vaccines. Current H5N1 stockpiles are made using the same process as seasonal vaccines, meaning that they do not need to undergo animal studies before going into humans.